# Identifying signatures of past and present cryovolcanism on Europa

Elodie Lesage [1] ✉, Samuel M. Howell [1], Marc Neveu [2,3], Julia W. Miller[1,4], Mariam Naseem [5], Mohit Melwani Daswani [1], Justine Villette [6] & Steven D. Vance [1]

Europa, the most visibly active icy moon of Jupiter, is a prime target for the search for life in the outer solar system. Two spacecraft missions, Europa Clipper from the National Aeronautics and Space Administration (NASA) and the Jupiter Icy Moon Explorer (JUICE) from the European Space Agency (ESA), will observe its surface, probe its interior structure, and characterize the space environment starting in 2030. Occasional eruptions of water sourced from Europa's interior may provide a window on the interior conditions and habitability of the moon. Here, we investigate the storage and evolution of briny water in Europa's ice shell and propose a framework to interpret spectral, thermal, radar and gravity data collected by future missions. We show that it is possible to discriminate between water erupting from the deep ocean or from shallow liquid reservoirs using combined measurements of the material's salinity, surface temperature and ice shell thickness.

The eruption of briny solutions on the surfaces of icy ocean worlds has been proposed to explain the origin of vapor plumes, domes, dark deposits, and lobate, flow-like features[1–9]. Additional surface expressions such as chaos, lenticulae, double ridges, and central pits in impact craters are plausibly explained by liquid brine ("cryomagma") reservoirs in Europa's shallow ice shell[10–15], despite challenges related to their formation and retention[16–19]. Subsurface liquid reservoirs may thus store water and non-water material and provide a mechanism for their transport through Europa's ice shell.

Cryomagma freezing is expected to pressurize reservoirs, potentially triggering eruptions[2,7,20,21]. Potential erupted flows or plumes, if observed by upcoming missions, could provide new insights into the habitability and evolution of Europa's ice shell by revealing properties of transient and long-lived interior reservoirs of liquid water environments[6–8,21–27]. The identification and characterization of erupted material and shallow subsurface liquid reservoirs would therefore be of great interest to understand Europa's habitability and, in the distant future, to access and analyze subsurface water[28–33].

However, even if ongoing eruptions are observed directly, the composition of erupted material may not represent the composition of the reservoir at its time of formation, because melting and freezing of trapped brines are expected to affect their chemical composition through salt rejection, entrapment, precipitation, and dissolution[22,25,34,35]. Characterization of source reservoirs requires understanding how observed physico-chemical properties of surface features are affected by subsurface reservoir evolution processes, ambient ice temperature, as well as their depth within the ~ 20–40 km ice shell[36]. In previous studies, Naseem et al.[22] characterized the compositional evolution of brines of varying starting compositions relevant to Europa's ocean[37] during freezing of a cryoreservoir. Lesage et al.[21] modeled reservoir rheology, pressurization and fracturing. Finally, Howell (2021)[36] presented a framework to model ice shells of time-variable thickness.

In this work, we bridge possible surface expressions of the erupted material and the physical signatures of source reservoirs with their physicochemical properties. We develop a novel program, CRYOLA-VASAURUS, publicly available at https://github.com/ElodieLesage/Cryolavasaurus[38]. Within this program, we self-consistently simulate

[1]Jet Propulsion Laboratory, California Institute of Technology, Pasadena, CA, USA. [2]Department of Astronomy, University of Maryland, College Park, MD, USA. [3]Planetary Environments Laboratory, NASA Goddard Space Flight Center, Greenbelt, MD, USA. [4]Department of Earth, Environmental, and Planetary Sciences, Brown University, Providence, RI, USA. [5]Department of Geology, University of Maryland, College Park, MD, USA. [6]Nantes Université, Univ Angers, Le Mans Université, CNRS, Laboratoire de Planétologie et Géosciences, LPG UMR 6112, 44000 Nantes, France. ✉e-mail: elodie.lesage@jpl.nasa.gov

the interplay of physical and chemical processes driving the freezing and eruption of shallow subsurface cryomagma reservoirs on Europa (see Methods, Section 1). We model an ice shell of time-variable thickness in which is embedded a shallow liquid reservoir of time-variable size and composition (see Methods, Section 2). The thermal, physical and compositional evolution of the ice shell and the reservoir, including convection in the ice shell and phase change at all interfaces, are computed simultaneously. We model heat transfer through a 1-dimensional (1D), spherically symmetric grid using an explicit finite-difference approach forward in time. Due to our 1D approximation, the reservoir is implemented as a layer of liquid water of imposed initial thickness and depth to the top of the layer. We track through time the ice shell temperature and thickness and the reservoir temperature, composition, and frozen fraction. We predict the characteristics of the erupted liquid at the time of each eruption, depending on the initial composition, depth, and thickness of the reservoir. Details on the numerical simulation are given in the Supplementary Model Description.

## Results

### Reservoir signatures and effect of salinity

Subsurface cryoreservoirs may leave several observable hints of their current or recent presence within the ice shell, including surface composition changes and thermal anomalies (Fig. 1a and b).

Continuous reservoir freezing induces overpressure, triggering repeated eruptions until no liquid remains or until pressure is relieved by viscous relaxation in the icy shell. Reservoirs 5 km thick (the thickest tested) remain active for up to 60 kyrs and trigger up to a few hundred eruptions, depending on their volume and depth. For a narrative explanation of the results, we chose to use a nominal reservoir that is 1 km thick and 1 km below the surface. For this reservoir, we obtain a total erupted volume of $4.5 \cdot 10^{-2}$ km$^3$ at the end of its 15 kyr active life (Fig. 1a), which represents an eruption of about 8% of the initial volume. The erupted volume is approximately the difference between the densities of ice and liquid water multiplied by the initial volume of the reservoir, which is expected for the assumptions we made: (i) the reservoir is disconnected from the ocean with no water refilling, (ii) the frozen ice layer has no porosity, keeping the water from infiltrating the ice shell, and (iii) the cryomagma is maintained at a pressure high enough that it does not volatilize before it reaches the surface.

For the nominal 1 km thick reservoir located 1 km below the surface, the surface thermal anomaly (Fig. 1b) resulting from the conduction and diffusion of the heat and latent heat from the reservoir peaks at +1.5 K after 20 kyr have elapsed since reservoir emplacement. The propagation of this heat in the ice shell also induces a local thinning below the reservoir (Fig. 1c). The thinning is insignificant for the shallow, nominal reservoir (<3 meters), but increases with reservoir

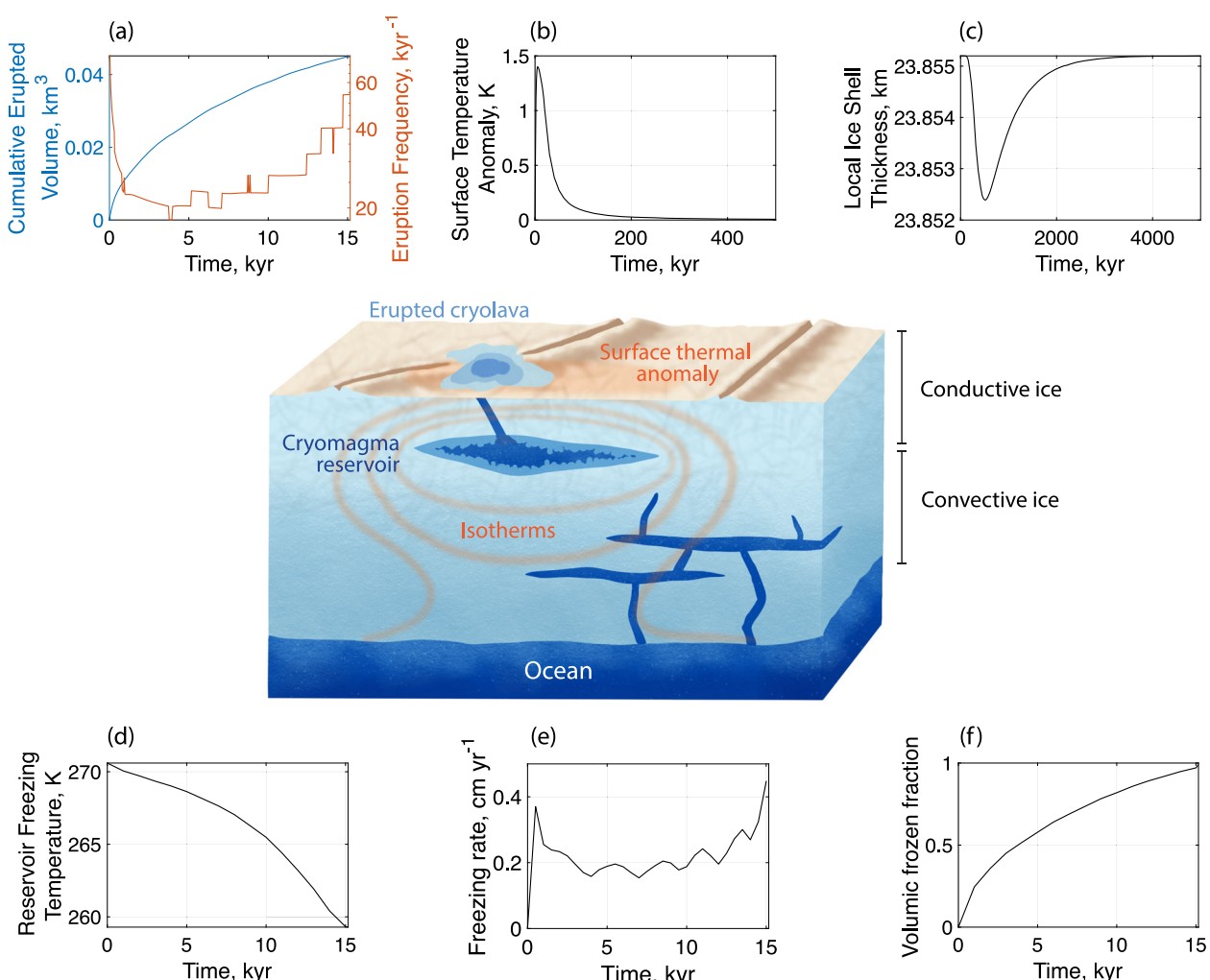

**Fig. 1 | Observable signatures and freezing characteristics for the 1 km thick reservoir located 1 km deep below Europa's surface. a** Cumulative erupted volume, **b** surface temperature anomaly, **c** local ice shell thickness, **d** cryomagma freezing temperature, **e** freezing rate, and (**f**) volumic frozen fraction. Time (*x*-axis) scales differ between panels. The emplacement mechanism of the source reservoir is beyond the scope of this study.

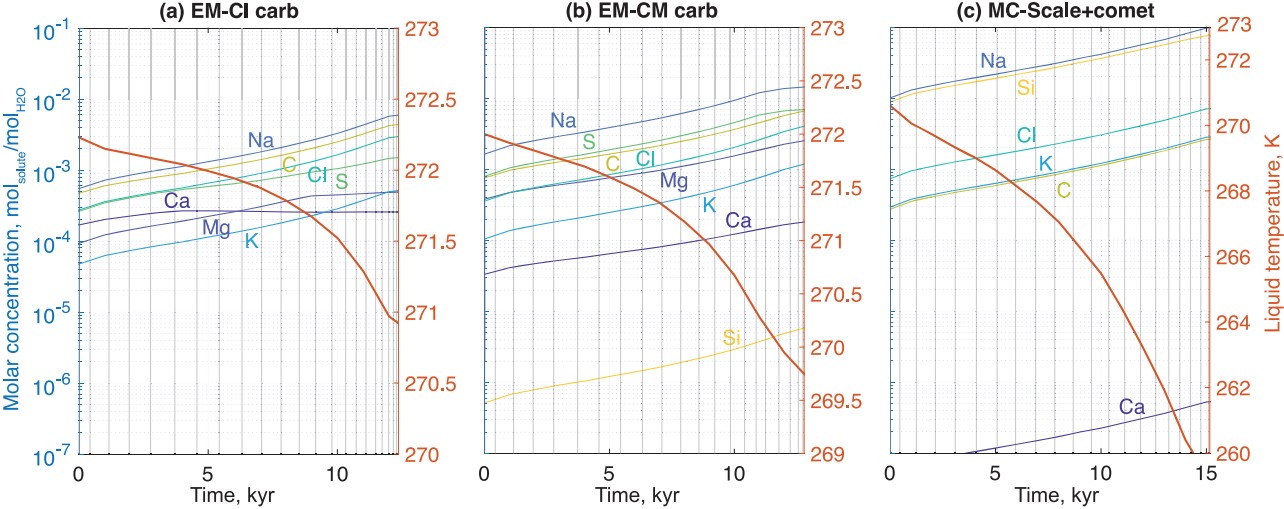

**Fig. 2 | Evolution of the composition of a freezing 1 km thick, 1 km deep reservoir through time.** Here we show reservoirs injected from three different plausible ocean compositions corresponding to three different primordial Europa compositions[22,37]: **a** CI chondrites, **b** CM chondrites, and **c** a mix of chondrites scaled to distance to the Sun with the addition of cometary material. Salts present in very low concentration are not represented on the plots (see Methods for more information on input salt molalities). 20 eruptions occur between each vertical gray line.

depth. In particular, thinning is strongest ~ 500 kyr after the onset of emplacement. Freezing is delayed due to the timescale of heat propagation by a factor of 30 relative to the freezing time of the reservoir. The thinning of the ice shell is also a long-lived signature, retaining 50% of its maximum value after >1 Myr.

In our simulations, freezing occurs slowly enough (Fig. 1d–f) that, combined with efficient mixing in the reservoir (see Methods, Section 3), it likely takes place at chemical equilibrium, meaning that the salts rejected in the liquid phase continue to participate in chemistry. This assumption is consistent with experimental data for terrestrial seawater and previous Europa literature[39]. This remains true for all reservoir sizes (0.2–5 km) and depths (1–15 km) simulated.

Notably, despite the antifreeze properties of dissolved salts that decrease the solidus temperature, cryomagma composition has only a minor effect on freezing time, freezing rate, eruption interval and total erupted volume. The nominal reservoir freezes at 11.4 kyr if it is composed of pure water and <15 kyr if it is briny, with freezing rates of <0.6 cm/yr in all cases. As a result, the induced surface thermal anomaly, and ice shell and lithosphere thinning do not inform on source composition, but rather on reservoir physical properties. The effect of salinity is minor because only ≈ 1/15th of the energy exchanged with the reservoir arises from the transfer of thermal energy, the bulk arising from the latent heat released by liquid water upon freezing. These results confirm previous predictions from fixed eutectic freezing models[7,21].

### Erupted composition

Simulated eruption intervals and reservoir compositions are shown in Fig. 2 for three initial liquid compositions. Current-day oceanic compositions are predicted by rocky mantle devolatilization and water-rock interaction simulations of candidate Europa interiors; they depend on the assumed composition of material accreted during Europa's formation[37]. We consider two early formation scenarios in CI (Fig. 2a, "CI") or CM chondrite-rich environments (Fig. 2b, "CM") and one formation scenario in mixed chondrites and water ice-depleted environment with subsequent addition of cometary material (Fig. 2c, "MC-Scale+comet"). Total dissolved solid concentrations for these formation scenarios are respectively 4.8, 10.5, and 45 g/kg, dominated by dissolved silica in the MC-Scale+Comet scenario. More information on the input compositions is given in Methods.

Importantly, despite the total salinity increasing during freezing, we find that relative proportions of the different salts in the erupted material are very similar to those in the initial reservoir, suggesting that determination of compositions of erupted material would provide meaningful insight into the composition of parent subsurface liquids. We also observe that for all three tested brine compositions, efficient salt rejection during freezing results in a gradual increase in solution salinity by an order of magnitude throughout the lifetime of the reservoir. This piece of information represents a first step in discriminating between eruptions sourced from the deep ocean or from evolved, local reservoirs. However, we do not address here how the composition and structure of the material are modified on the surface during solid-vapor phase changes or irradiation[40–42], which are discussed in the following.

### Influence of reservoir size and depth

Reservoir size and depth have a determining role in surface signatures (Fig. 3). The surface temperature anomaly above the reservoirs, assumed here to be of the same spatial extent as the reservoir itself, mainly depends on the reservoir depth. We obtain a maximum anomaly value of ~1.5 K for the shallowest reservoirs simulated (Fig. 3a). Variations in ice shell thickness are more pronounced with reservoir thickness and depth (Fig. 3b). This result is intuitive, as larger reservoirs release more energy stored in the liquid phase of water, while deeper reservoirs more rapidly and efficiently transport heat to the ice-ocean interface. Variations in ice shell thickness range from <1 m for reservoirs <1 km thick and 1 km deep, to 100 m for reservoirs >5 km thick and 5 km deep. Finally, the total amount of fluid that erupted from a reservoir depends on its initial volume and the density difference between the liquid and solid phases, but also on the temperature of the ice shell surrounding it. As the ice shell temperature increases, viscous relaxation becomes more efficient and prevents the reservoir internal pressure from building up, preventing the pressurization of reservoirs from achieving the critical stress required for eruption. Figure 3c shows the erupted volumes assuming a spherical reservoir. In our simulations, the deepest reservoirs (>10 km) never trigger eruptions for reservoir thicknesses >200 m.

Furthermore, we show that deeper reservoirs, located in warmer ice, freeze more slowly because the thermal gradients driving

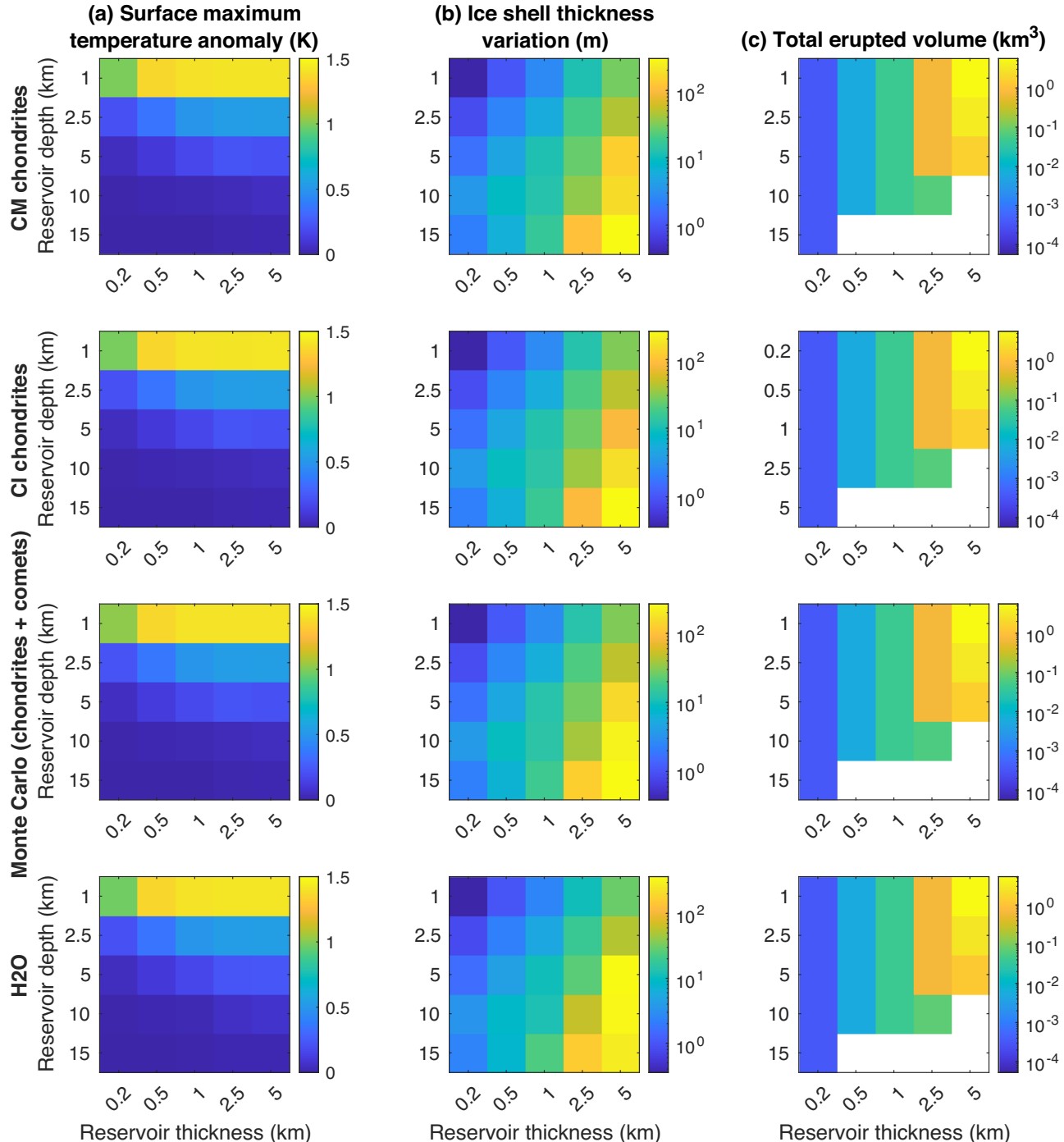

**Fig. 3 | Effect of reservoir thickness and depth on various signatures. a** Surface temperature anomaly, **b** ice shell thickness variation, and (**c**) total erupted volume. Blank cells in (**c**) indicate reservoirs that do not trigger eruptions.

heat conduction are smaller, lengthening the interval between eruptions. As a result, the eruption frequency decreases with the depth of the reservoir. The local onset and/or enhancement of solid-state convection develops faster below deeper reservoirs, and convection cells take less time to reach the base of the ice shell. As a result of the exponential decrease in the viscosity of the ice with temperature, the active reservoir lifetime decreases with increasing reservoir depth. This might seem counter-intuitive as deeper, warmer reservoirs freeze at slower rates, but this finding is explained by the fact that eruptions cannot be triggered in ice that dissipates accumulating stress viscously at or faster than the freezing timescale[21].

## Discussion

We find that the composition of the erupted material reflects proportionally the composition at the reservoir formation time. Thus, a comparison between the surface composition at a candidate cryovolcanic site and planned independent measurements of the salinity of Europa's ocean[33,43,44] will contribute evidence as to whether the parent reservoir liquid is sourced from the ocean or is instead due to melting of ice shell material. Eutectic mixtures melted in situ would refreeze near the eutectic, and the material stored in the reservoir and erupted at the surface would thus be relatively enriched in lower-eutectic-temperature salts, such as chlorides, compared to the oceanic compositions tested. For example, chloride enrichment has been reported

from Earth-based observations at higher concentrations in dark terrains of the trailing hemisphere[45].

Our findings also show that cryomagma salinity increases by an order of magnitude between reservoir emplacement and total freezing. Thus, regions identified as candidate cryovolcanic sites may show gradational spatial variations in surface composition associated with the eruptive sequence. These changes in surface properties may be detectable by the Europa Clipper mission's suite of optical remote sensing instruments comprising the Europa Imaging System (EIS)[46] and the Mapping Imaging Spectrometer for Europa (MISE)[35]. The MISE camera is dedicated to the detection and mapping of non-ice materials (salts, acids, organics) and has a spatial resolution of up to to 25 m/px[47].

We note that inference of reservoir composition from observations of erupted brines is limited by the following caveat: several processes can affect the composition and structure of ice grains formed from the erupted material. Detection of frozen brines and measurement of their composition is complicated by processes occurring during solidification, such as the vitrification of grains[41]. Enrichment in certain species and depletion in others also occur during phase changes, as demonstrated by laboratory work. Enrichment in NaCl preferentially to $Na_2CO_3$ has been observed under conditions relevant to Enceladus and Ceres[42]. Enrichment in $Na_2SO_4$ and $MgCl_2$ was also observed under Europa conditions[48]. Finally, sublimation and irradiation of frozen brines also affect their composition on longer time scales, as shown by[40] for frozen chloride brines under Europa conditions. In general, a disconnect is often observed between source compositions and frozen brines, comprising an important and ongoing area of laboratory work and the development of compositional models[41,42,49,50]. Potential opportunities to directly sample freshly erupted plume material[33,35,43] may be key in resolving the uncertainties introduced by exposure of cryomagma to space.

Although this study did not model the detection of any predicted signature presented here, these may be detectable by several instruments aboard the Europa Clipper spacecraft. Short-lived thermal anomalies as found here are plausibly detectable by the Europa Thermal Emission Imaging system (E-THEMIS), which has an expected temperature precision of 1.2 K[33,43,51,52]. Europa Clipper's radar instrument REASON may also be able to detect subsurface liquid brine or salt-rich layers[53], as well as more subtle changes in the depths at which thermally diagnostic interfaces occur (e.g. pore curing or liquid brines) using sparse echoes[54]. Finally, density changes associated with the presence of a large-scale liquid reservoir and any additional local thinning of the ice shell might result in a gravity anomaly detectable by gravity and radio science[44].

In conclusion, the combined measurements of thermal, spectral, radar and gravity anomalies may enable the identification of current or recent (0 - ~ $10^2$ kyr old) reservoir emplacement and freezing. This identification could constrain the solution space comprising the timing of emplacement, reservoir depth, and initial reservoir volume. Our simulations and results demonstrate how the upcoming Europa Clipper and JUICE missions will investigate problems central to the evolution and habitability of the Galilean moons through multidisciplinary measurements and their synergy[33,43,51,55–58].

## Methods
### Model principle
CRYOLAVASAURUS simulates the thermal, mechanical, compositional, and temporal evolution of the ice shell and cryomagma reservoir by solving for the conservation of enthalpy. It uses finite differences in a one-dimensional (1D) spherically symmetric shell, propagated explicitly forward in time. The surface temperature, the thickness of the ice shell and its thermal structure, and the reservoir material phases and compositions are determined self-consistently by the simulation. Extensive documentation on the CRYOLAVASAURUS

framework and model can be found in Supplementary Model Description.

The thermal solver described in Supplementary Information, Section 1, includes:
- the exchange of energy due to temperature differences through conduction, diffusion, and convection;
- the exchange of latent heat during melting and freezing at the ice-reservoir and ice-ocean interfaces;
- the release or capture of energy due to changes in the specific enthalpy of formation of brines and hydrated salts;
- the addition of radiogenic heating from Europa's rocky interior at the seafloor;
- the rheology-dependent viscous dissipation of gravitational potential energy within the ice shell due to diurnal eccentricity tides;
- the conservative balance of energy associated with insolation, Europa's black body radiation, ice sublimation, and the escape of heat from Europa's icy surface.

Within this framework, we account for the following ways in which conservative energy transfer can affect the system:
- *Mechanical energy*: Pressure and volume change, allowing useful work to be done through deformation (e.g. fracturing) and mass transport (which fundamentally affects heat transport through advection).
- *Rate of mechanical change*: The mechanical properties of the material change (e.g. compressibility, elastic moduli, Lamé parameters, sound speed, rheology), affecting the future partitioning of energy storage and dissipation.
- *Thermal energy*: The temperature changes according to the thermal properties of the material and its proximity to phase boundaries in both the state of matter and the composition.
- *Composition*: Mineralogy may change, both in abundance and in the creation and destruction of some minerals, according to the specific enthalpies of formation of the possible materials, their physical and thermal properties, and changes in stability of materials as pressure and temperature evolve.
- *Phase*: Materials can change their state of matter according to the pressure and temperature evolution of their surroundings with respect to their phase space.
- *Rates of thermal and compositional change*: The thermal properties of the material change, controlling how much the next Joule of energy will alter the temperature, phase, and composition.

### Boundary and Initial conditions
In this section, we summarize the conditions imposed at key interfaces within the simulation.

**Ice shell properties.** The temperature at the ice-ocean interface is set as the melting temperature of the oceanic material. At the surface (ice-space interface), the temperature is self-consistently calculated at each time step using a heat flux balance. This balance accounts for radiogenic heat flux from Europa's interior, solar insolation heat flux, sublimation heat flux, and radiative heat flux. Details on this calculation are given in the Supplementary Model Description (Section 1). The thickness of the ice shell is not imposed, but is determined self-consistently by the simulation. Models are initiated with a conductive temperature profile near the equilibrium thickness and then allowed to evolve thermally for 5 Myr prior to reservoir emplacement so that a steady-state ice shell structure is ensured.

**Reservoir emplacement.** Whether it occurs rapidly by injection into the ocean or more slowly by in situ melting, reservoir emplacement must happen much faster than the timescale for the reservoir to freeze[16,17,59]. Here, we consider that the reservoir is emplaced

**Table 1 | Molalities (mol element per kg of H₂O) of the three input compositions used in this study**

|     | CI | CM | MC-Scale+Comet |
|-----|-----|-----|-----|
| Ca | $9.39 \times 10^{-3}$ | $1.86 \times 10^{-3}$ | $3.02 \times 10^{-6}$ |
| Mg | $5.16 \times 10^{-3}$ | $2.12 \times 10^{-2}$ | 0 |
| Na | $3.05 \times 10^{-1}$ | $9.19 \times 10^{-2}$ | $5.59 \times 10^{-1}$ |
| K | $2.63 \times 10^{-3}$ | $5.84 \times 10^{-3}$ | $1.62 \times 10^{-2}$ |
| Cl | $1.52 \times 10^{-2}$ | $2.00 \times 10^{-2}$ | $4.07 \times 10^{-2}$ |
| S | $1.46 \times 10^{-2}$ | $4.47 \times 10^{-2}$ | 0 |
| C | $2.63 \times 10^{-2}$ | $4.24 \times 10^{-2}$ | $1.51 \times 10^{-2}$ |
| Si | 0 | $2.86 \times 10^{-5}$ | $4.89 \times 10^{-1}$ |

Compositional data from[22,37].

instantaneously within the ice shell once the ice shell temperature reaches equilibrium to avoid the practical implications of simultaneously resolving numerous questions regarding reservoir formation and emplacement, and to maintain the focus of this study on reservoir evolution and detection. Because of our 1D approximation for the model grid, the liquid reservoir does not technically own a horizontal extent, but is rather described by a global layer of water of imposed initial thickness. However, we use a spherical description to better simulate energy transfer in and around the reservoir. We interpolate heat loss at the surface and base of the reservoir around the shell perimeter, conservatively scaled by the far-field ice temperature. This is a good approximation with only small uncertainties introduced through *de minimis* heat transport radially along the thin, nearly isothermal boundary normal to the horizontal plane.

**Initial cryomagma composition.** We use three plausible compositions for Europa's ocean, as well as a reference model with pure water. Melwani Daswani et al.[37] modeled the water-rock interaction for a young Europa to infer plausible present-day compositions of the ocean. The resulting ocean composition depends on the environment during Europa's accretion, e.g. the composition of chondrites and the bulk amount of available water. As detailed in Melwani Daswani et al.[37], the oceanic composition "CI" refers to a formation scenario in an environment rich in CI chondrites, and the composition "CM" in an environment rich in CM chondrites. The "MC-Scale+comet" composition was obtained using Monte Carlo simulations where the probability of accreting different types of carbonaceous chondrites (CI, CM, CV, CK, CR, and CO) was weighted by their formation distance from Jupiter. Cometary material was then added to compensate for insufficient volatiles produced by this accretion scenario[60], using comet 67P as a reference. These fundamentally different formation models explain the order-of-magnitude differences in elemental concentrations such as Si and Ca between scenarios. Table 1 summarizes the total element molalities for the three compositions used in the present study.

## Ice shell and reservoir thermal evolution

The thicknesses of the reservoir and ice shell evolve over time to account for phase-related enthalpy exchange at all interfaces. That is, the top and bottom reservoir boundaries and the ice-ocean interface move over time as a result of freezing and melting.

At the surface, the temperature evolves self-consistently by balancing incoming diurnally averaged equatorial insolation, $\dot{q}_{sol}$, and geologic heat flow through the surface, $\dot{q}_{geo}$, with heat losses due to blackbody radiation, $\dot{q}_{bb}$, and sublimation to maintain a rarefied atmosphere, $\dot{q}_{sub}$:

$$\dot{q}_{sol} + \dot{q}_{geo} = \dot{q}_{bb} + \dot{q}_{sub}. \tag{1}$$

See the Supplementary Model Description (Section 2) for further information.

We assume that the cryomagma is convecting during freezing, because the estimated Rayleigh number in the reservoir exceeds values that typically indicate convection ($Ra > 10^6 - 10^8$). For physical parameters on the order of those used in this study ($\rho = 1000$ kg m$^{-1}$ the liquid density, $g = 1.35$ m s$^{-2}$ Europa's gravity, $\alpha = 10^{-3}$ K$^{-1}$ the water thermal expansivity, $cp = 4.2$ kJ kg$^{-1}$ the liquid water specific heat capacity, $\eta = 10^{-3}$ Pa s the liquid water viscosity, $k = 0.5$ Wm$^{-1}$K$^{-1}$ the liquid water thermal conductivity; see[61]), and with a realistic maximum temperature gradient of 1 K across the reservoir and a characteristic reservoir vertical extent of 2 km, we get $Ra \simeq 10^{19}$. Vigorous convection implies that heat is efficiently transported in the liquid phase, and thus we consider it isothermal.

Finally, we model phase changes at the reservoir interfaces and the ice-ocean interface. We do so by tracking the energy flux across these boundaries (see Section 4 below). In the reservoir, we use the freezing paths from Naseem et al.[22] (Section 4) to estimate the amount of cryomagma that freezes given the energy exchanged at the interfaces at each time step. We update the temperature solution of the boundary elements consistently with the compositional evolution of reservoir and icy material, and freezing, using the above-referenced simulation to partition changes in energy. We track the interface locations and update all physical parameters. The process we use to calculate the effective thermal conductivity at interface nodes is described in the Supplementary Model Description, Section 1.

## Cryomagma chemical evolution during freezing

Naseem et al.[22] modeled the freezing of inclusions of ocean material in the ice shell using the oceanic compositions predicted by Melwani Daswani et al.[37] as inputs. In Naseem et al.[22], compositions are treated as a function of decreasing temperature[22] and obtained using PHREEQC v3[62] and the thermodynamic data converted by Toner et al.[63] from previous work[34,64]. The model in Naseem et al.[22] calculates the chemical sequence of forming species with decreasing temperatures, while ignoring the time evolution of the system temperatures. The link to the data spreadsheets used in this study, as well as information on their labeling, are given in the Supplementary Model Description, Section 3.

The freezing rates we obtain are under 1 cm/yr, indicating a slow freezing scenario in which salts are efficiently rejected from the formed ice and concentrate in the liquid phase, where the salinity increases over time. Because convection is efficient (Section 3), precipitated salts are assumed to remain suspended in the liquid phase and still participate in chemical reactions (equilibrium crystallization). This end-member case was simulated in[22] and we use the corresponding data as input for our model.

We do not account for the effect of phase change-induced pressure changes on compositions, which is expected to be negligible[22]. Liquid density is an output of PHREEQC and is used to calculate the overpressure in the reservoir. Density change due to compression is calculated and used to determine if an eruption is triggered at each time step (Section 5).

To bridge the PHREEQC outputs of composition as a function of decreasing temperature with the computed reservoir thermal evolution, we track energy removed from the reservoir in both models. Our simulation uses time steps, and the PHREEQC data use temperature steps, but we can reconcile these by expressing both models in terms of energy steps. The initial total amount of energy stored in the reservoir can be calculated using the liquid temperature $T_{liq}$:

$$E_{init} = \rho_{liq} \, cp \, T_{liq}. \tag{2}$$

We then track how much energy is removed from the reservoir at each time step of the simulation and subtract it from $E_{init}$. In parallel, we calculate how much energy is left in the reservoir for each PHREEQC

data point using the changes in temperature and amount of liquid and solid materials in the PHREEQC output files. Using interpolations between the PHREEQC data points, we retrieve the composition associated with the energy state of the reservoir at each time step. More details about this process are available in the Supplementary Model Description, Section 4.

### Reservoir pressurization and eruption

We further develop the framework of Lesage et al.[21] to track reservoir pressurization and eruption. Reservoir pressurization arises from the increase in water volume after freezing, with volume expansion opposed by the incompressibility of the melt. When the stress on the reservoir wall exceeds a threshold value, determined as function of the ice tensile strength, a fracture propagates to the surface and an eruption is triggered.

We self-consistently evaluate the accumulation of elastic stresses due to pressurization of the compressible medium, the viscoelastic relation of those stresses with time according to the rheology, and the ability of the system to trigger a new eruption mechanically. Details on the model implementation of these processes are included in the Supplementary Model Description, Section 5.

### Data availability

The data generated in this study have been deposited in the public GitHub repository https://github.com/ElodieLesage/Cryolavasaurus/tree/main/Onput and permanently archived at https://doi.org/10.5281/zenodo.14768339. The compositional data from Naseem et al.[22], as well as the input files used to run the CRYOLAVASAURUS simulation, are available at https://github.com/ElodieLesage/Cryolavasaurus/tree/main/Input and permanently archived at https://doi.org/10.5281/zenodo.14768339. These files contain the input parameters used to obtain all the results presented in this study. All input and output data is fully available at the links above and downloadable without restrictions.

### Code availability

The compositional data showed in Fig. 2 was obtained using the PHREEQC routine available at https://github.com/MarcNeveu/frezchem and following the methods in Naseem et al.[22]. The CRYO-LAVASAURUS software used to obtain the results of this study is open-source and fully available at https://github.com/ElodieLesage/Cryolavasaurus, and archived at https://doi.org/10.5281/zenodo.14768339.

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

## Acknowledgements

Portions of this research were carried out at the Jet Propulsion Laboratory, California Institute of Technology, under contract with the National Aeronautics and Space Administration (NASA). This work was primarily supported by NASA's Solar System Workings program grants #80NM0018F0612 (E.L., S.M.H. and J.W.M.) and #80NSSC20K0139 (E.L., S.M.H., M.Ne., M.Na., M.M.D., S.D.V.). Parts of this work were also supported by NASA under award #80GSFC24M0006 (M.Ne., M.Na., S.D.V.) and through the Precursor Science Investigations for Europa program grant #80NM0018F0612 (S.D.V).

## Author contributions

E.L., S.M.H. and J.W.M. developed the CRYOLAVASAURUS numerical model and ran the simulations. M.Ne. and M.Na. developed the PHREEQC routine and provided the composition data. E.L., S.M.H., M.Ne., M.Na., J.W.M., M.M.D., J.V., and S.D.V. discussed the findings of the study and contributed to the final manuscript.

## Competing interests

The authors declare no competing interests.
