## [Transparent Peer Review file · Nature Communications]

Identifying signatures of past and present cryovolcanism on Europa

Corresponding Author: Dr Elodie Lesage

Version 0:

Reviewer comments:

Reviewer #1

(Remarks to the Author)

I have reviewed the manuscript "Signatures of past and present cryovolcanism on Europa: Composition, Geology, Interior" by Lesage and colleagues, submitted for publication in Nature Communications. The paper hypothesizes that a comparison between surface composition at a candidate cryovolcanic site and planned independent measurements of the salinity of Europa's ocean can be used to determine whether the parent reservoir liquid is itself sourced from the ocean or instead due to melting of ice shell material. This can be tested by the upcoming Europa Clipper mission, by looking for changes in surface properties. Regions identified as candidate cryovolcanic sites may show gradational spatial variations in surface composition associated with the eruptive sequence. The authors describe a physical and chemical model of freezing and eruption, and proposes a method to discriminate between oceanic or shallow reservoirs as the source for erupted material.

Below, please find my specific comments on the manuscript.

1. Line 103. "assume" rather than "presume."
2. Line 117. "observable hints" of cryoreservoirs are mentioned here, but would be good to describe them in some detail.
3. Line 125. The text states that the "The erupted volume is intuitively approximately the difference between the densities of ice and liquid water multiplied by the initial reservoir volume," but it's not necessarily that intuitive. Pressurization of the ocean might result in retention of the material in the subsurface. Moreover, depending on the strength of the ice, the entire shell could expand, potentially accommodating some or all of the excess volume without necessarily eruption. Finally, in the event of an eruption, a fair amount of the material will be converted to vapor and lost entirely. This might be counted as part of the erupted volume, but would be distinct from cryomagma.
4. Line 130. How detectable is a thermal anomaly of 1.5 K on the scale of a cryomagmatic feature? Could this be observed from orbit?
5. Figure 1. In the cartoon figure shown in the center, there are a number of sills, dikes, and other fluid filled cracks shown in the lower part of the ice shell. This part of the ice is warm and potentially convective. How are these cracks maintained against the flow of warm, ductile ice?
6. Line 176. Looks like there is an errant numeral '2' after "briny." This might be a citation that just needs brackets?
7. Lines 178–182. The discussion of the relatively low effect of salinity is very well described here and put into good context in terms of energy.
8. Figure 2 and Lines 205–210. I'm not sure I understand how chondritic compositions would match the ocean. Is it that the bulk Europa composition is taken to be chondritic and then the volatiles are separated out through differentiation?
9. Line 211. What is "a majority of dissolved silica" in this context. Silica is not super soluble in water, so I'd have thought that would be a relatively low number.

10. Line 304. Some discussion of the cameras on Europa Clipper and their ability to detect changes in surface properties as described here would be helpful.

11. Line 326. What is the resolution of the thermal anomalies detectable by E-THEMIS? Can it see anomalies at the ~1 K level over the spatial scales needed here?

12. Lines 448–449. The paper states that radial heat transport is not significant because the reservoir boundary is nearly isothermal. But I would think that would mean that lateral heat transport is small. Radial heat transport would be into and out of the reservoir? And that could still be important?

13. Line 446. The word “is” should be omitted.

14. Lines 476–481. The parameter values all appear reasonable, but some references should be given. Perhaps to put these in a table would be the easiest way to present this?

(Remarks on code availability)

I was able to access the code, but did not actually test it. The README file appears to be empty.

Reviewer #2

(Remarks to the Author)

Key results:

The authors report on a modeling study that investigates the storage, physicochemical evolution, and eruption of brines contained within discrete reservoirs within Europa’s ice shell. This work builds on previous work by the authors, and takes the important step of combining focus on the freezing reservoir itself with assessment of the composition of cryolavas that would be erupted, and the effect of the presence of the brine reservoir on the thickness of the ice shell. The authors present results for reservoir longevity, surface temperature anomalies, ice shell thickness variations, and total erupted volumes for three plausible ocean compositions, as well as pure water. They show that predicted erupted compositions bear close similarity to the initial reservoir/ocean composition, with increasing salinity over time as the reservoirs freeze, which would allow for discrimination between eruptions sourced from a subsurface reservoir vs. the ocean. As such, this is a very comprehensive treatment that provides predictions that can be tested with a range of instruments flying on the upcoming missions to the Europa system.

Significance:

The work provides a set of predictions to inform upcoming missions to the Jupiter System that will allow for identification of past or present fluid reservoirs and their eruptions. Identification of locations hosting liquid water is highly significant for the outer solar system exploration objective of searching for habitable environments.

Data, methodology, validity of approach:

As we await new data sets from upcoming missions to Europa, we rely heavily on theoretical modeling, in this case modeling of the physicochemical evolution and consequences of brine reservoirs in the ice shell. The description of the modeling components, in both the main text and supplemental material, is sufficiently detailed, and the approach adopted seems to be robust. The strength of the present work lies in combining different model components (conservation of heat through the ice shell in response to the freezing or the brine reservoir, chemical evolution of the reservoir, pressurization and eruption of the brine reservoir, and stress response of the ice shell), in order to produce a more holistic evaluation of the dynamics of brine reservoirs in the ice shell. The authors explore an appropriately broad parameter space, which captures the plausible (but as yet unknown) range of conditions that may exist. Figure 3 provides an excellent summary of the results of all simulations. While we still don’t know how such liquid reservoirs might be emplaced in the first place (which the authors acknowledge), I concur that the conclusions regarding reservoir and eruptive evolution are valid and well supported by the modeling.

This is a modeling investigation that uses previously published data as model parameters and constraints. As such, those data have already been validated. The code underpinning this work is provided on GitHub, but has not been test-run by this reviewer.

Clarity and context: The manuscript was a pleasure to read. It is well structured and clearly written, and appropriately discusses previous work as context for the new results presented.

Suggested improvements:

The quality of the manuscript is high and I have only minor comments, marked up on the manuscript and supplemental material files. In addition to typo, formatting, and wording comments left solely in the marked-up files, I reiterate a few comments (all minor) here:

- Line 130. “phase-energy” I assume you mean phase-change energy here (i.e. latent heat)?
- Line 132. If I’m reading figure 1c correctly, it looks like ~3 m thinning, not 30 m.
- Line 220. Add “at the surface” after “modified” for clarity
- Line 280. Regarding “... because the thermal gradients driving heat advection are lower...”, do you mean heat conduction through the ice, or not just heat conduction, but also advection due to ice convection? In either case, conduction is relevant

so should be included.

– Line 295. I agree that ocean-derived liquids would like differ in composition than liquids derived from in situ heating. But what if the locally melted ice was salty? Chivers et al. have shown that complex compositional layering can be produced in subsurface ice, so the compositiona; doistinction between ocean-derived and locally-derived (in-situ) fluids might not be as clear cut as briny vs. pure water fluids.

– Line 424. Last line of paragraph is incomplete.

– Line 443-445. Description of reservoir geometry: It's hard to envision what you mean here. An annulus is the region between two concentric circles... ring-shaped like a washer. I don't see how a spherical reservoir can be represented by an annulus. Do you simply mean that the reservoir is represented in 2D by a circle rather than an annulus? A clearer explanation (or a figure) is needed to clarify the geometry.

– Line 479. Ra is dimensionless, so here do you mean that to calculate Ra you need to multiply by some temperature and the dimension of the reservoir? Please clarify by providing explicit explanation of K and D. Or perhaps give the Rayleigh number as $Ra = 5 \times 10^{10}(\Delta T D^3)$, where ΔT is... and D is...

References.

The referencing is quite comprehensive but the authors might consider whether they would like to cite the following recent papers, which also bear upon the issues discussed in the paper:

– Chivers et al. (2023) Stable brine layers beneath Europa's chaos, PSJ

– Carnahan et al. (2022) Surface-to-ocean exchange by the sinking of impact-generated melt chambers on Europa, GRL

– Hesse et al. (2022) Downward oxidant transport through Europa's ice shell by density-driven brine percolation, GRL

– Winkenstern and Saur (2023) Detectability of local water reservoirs in Europa's surface later under consideration of coupled induction, JGR Planets

(Remarks on code availability)

I did view the GitHub site, but did not have sufficient time to dedicate to figuring out how to run the code. A more detailed readme file would be beneficial.

Reviewer #3

(Remarks to the Author)

This paper addresses an important topic for future explorations of Europa's habitability with NASA's Europa Clipper and ESA's JUICE missions, offering a comprehensive approach to distinguish the source reservoir of erupted water. The manuscript is generally well-written, and I hope my comments will help improve the clarity of the paper.

1. Lines 168-172: This is a reasonable assumption, given experimental data for terrestrial seawater. You could also mention that this assumption has been used in existing Europa literature, such as Zolotov & Shock (2001, JGR, 106, 32895-32827).

2. Ocean Composition Table: It would be helpful for readers to include a table describing the assumed ocean composition (i.e., key ions, dissolved volatiles, pH) used in these calculations, even though this information is cited from [40]. A brief description of the major differences in these cases would also be appreciated, as it's not immediately clear which comet compositions were adopted (e.g., is it from comet 67P or an average composition of multiple comets?).

3. Figure 2a & 2c: Fig. 2a seems to omit the evolution of Si, and Fig. 2c refers to a mix of CI and CM with added cometary materials, but S and Mg appear to be missing. Without further explanation, it's not obvious why Si and Ca differ by an order of magnitude between Fig. 2b and 2c, given that Fig. 2c is a mixture of materials from CI and CM, with some cometary input.

4. Volatile Exsolution: While volatile exsolution isn't the main focus of this paper, it's been discussed in ocean worlds literature as a potential driver for eruptions (e.g., Mitchell et al. 2024, <https://doi.org/10.1029/2023JE007977>). Within the context of this model, it might be useful to discuss to what extent this process might contribute to the eruption process.

5. Thermal Implications (Line 227): Given the maximum increase of 1.5K for the shallowest reservoirs, what are the implications for thermal measurements in identifying hotspots? Does this explain why telescopic measurements have not reliably detected endogenic heat? (e.g., <https://iopscience.iop.org/article/10.3847/1538-3881/aa8769>)

6. Testing Models with Current and Future Missions: The paper provides a strong discussion of how future missions could test these models. However, is there an opportunity for current or near-future Juno observations to help? While the discussion focuses on spectral, E-THEMIS, and REASON instruments, it's unclear how future gravity data could help distinguish between water erupted from deep ocean sources versus shallow liquid reservoirs, as mentioned earlier in the abstract.

(Remarks on code availability)

The code appears to organized, but the README file does not provide any instructions for installing and running the application

Version 1:

Reviewer comments:

Reviewer #1

(Remarks to the Author)

The authors have responded to all my comments on the original submission, and I'm ready to recommend this for publication. I have just a couple of further recommendations, which can be easily handled during the proofs stage; there is certainly no need for another round of review.

Regarding my original point 3, the authors explanation seems reasonable, but it would be good to include a mention of these complicating mechanisms (e.g., pressurization, expansion, vaporization), note that their not being included, and a brief (one sentence) explanation of why these can be safely ignored at this stage.

Regarding my original point 10, I see the authors cite the MISE paper in SSR, but not the corresponding EIS paper (Turtle et al., 2024). That one is now published online: <https://link.springer.com/article/10.1007/s11214-024-01115-9> and should surely be included here.

(Remarks on code availability)

I have not actually tried using the code, but have had no problems accessing it. The README and License are complete now.

Reviewer #3

(Remarks to the Author)

I have carefully reviewed the revised manuscript and the additional information provided by the authors. I am satisfied that my comments and concerns have been adequately addressed. The revisions improve the clarity and robustness of the study. I recommend the manuscript for publication in its current form.

(Remarks on code availability)

The authors thank sincerely all three reviewers for their encouraging reviews and appreciate the numerous, very helpful comments on the main manuscript, methods, and supplementary information. We detail hereafter how we addressed each comment specifically.

Reviewer #1 (Remarks to the Author):

I have reviewed the manuscript “Signatures of past and present cryovolcanism on Europa: Composition, Geology, Interior” by Lesage and colleagues, submitted for publication in Nature Communications. The paper hypothesizes that a comparison between surface composition at a candidate cryovolcanic site and planned independent measurements of the salinity of Europa’s ocean can be used to determine whether the parent reservoir liquid is itself sourced from the ocean or instead due to melting of ice shell material. This can be tested by the upcoming Europa Clipper mission, by looking for changes in surface properties. Regions identified as candidate cryovolcanic sites may show gradational spatial variations in surface composition associated with the eruptive sequence. The authors describe a physical and chemical model of freezing and eruption, and proposes a method to discriminate between oceanic or shallow reservoirs as the source for erupted material.

Below, please find my specific comments on the manuscript.

1. Line 103. “assume” rather than “presume.”

Done

2. Line 117. “observable hints” of cryoreservoirs are mentioned here, but would be good to describe them in some detail.

Added “including composition changes and thermal anomalies”

3. Line 125. The text states that the “The erupted volume is intuitively approximately the difference between the densities of ice and liquid water multiplied by the initial reservoir volume,” but it’s not necessarily that intuitive. Pressurization of the ocean might result in retention of the material in the subsurface. Moreover, depending on the strength of the ice, the entire shell could expand, potentially accommodating some or all of the excess volume without necessarily eruption. Finally, in the event of an eruption, a fair amount of the material will be converted to vapor and lost entirely. This might be counted as part of the erupted volume, but would be distinct from cryomagma.

These are all good points, although none of these mechanisms are accounted for in this study. Further modelling efforts would be necessary to include them. In the current state of the model, the erupted volume is mostly proportional to the ice-water density difference, modulo the reservoir viscous deformation. We removed “intuitively” to avoid over-simplification.

4. Line 130. How detectable is a thermal anomaly of 1.5 K on the scale of a cryomagmatic feature? Could this be observed from orbit?

We discuss this later in section “Implication for future missions”.

5. Figure 1. In the cartoon figure shown in the center, there are a number of sills, dikes, and other fluid filled cracks shown in the lower part of the ice shell. This part of the ice is warm and potentially convective. How are these cracks maintained against the flow of warm, ductile ice?

The thickness of the ductile ice layer is even more poorly constrained than the total ice shell thickness (e.g. Howell, 2021), and the exact mechanism from which sills are emplaced in the ice shell is unknown. From previous studies and the present one, we find that sills are more likely to be emplaced through a short-time mechanism as slower ones would result in downward drainage of brines. Sill intrusion through fractures thus seems a likely emplacement scenario despite the limitations of this mechanism, which we make explicit in the introduction. We added “The emplacement mechanism of the source reservoir is beyond the scope of this study.” in the figure caption.

6. Line 176. Looks like there is an errant numeral ‘2’ after “briny.” This might be a citation that just needs brackets?

Removed.

7. Lines 178–182. The discussion of the relatively low effect of salinity is very well described here and put into good context in terms of energy.

Thank you!

8. Figure 2 and Lines 205–210. I’m not sure I understand how chondritic compositions would match the ocean. Is it that the bulk Europa composition is taken to be chondritic and then the volatiles are separated out through differentiation?

Depending on the formation scenario for Europa, the rocky mantle may be enriched in CI or CM chondrites. The ocean composition is expected to reflect the mantle composition after billions of years of ocean-mantle chemical interaction (Melwani Daswani et al., 2021). We clarified by adding the following sentence: “Current-day oceanic compositions are predicted by devolatilization and water-rock simulations of candidate Europa interiors, which may vary depending on the conditions at Europa's formation”.

9. Line 211. What is “a majority of dissolved silica” in this context. Silica is not super soluble in water, so I’d have thought that would be a relatively low number.

Melwani Daswani et al. (2021) do find an important amount of dissolved silica in the ocean for the “MC-Scale + Comet” formation scenario (around 30+ g/kg). In this scenario, silica solubility is greatly enhanced by the ocean’s highly alkaline pH, due to high amount of dissolved Na. We rephrased to clarify: “Total dissolved solid concentrations for these formation scenarios are respectively 4.8, 10.5, and 45 g/kg, dominated by dissolved silica in the MC-Scale+Comet scenario.”

10. Line 304. Some discussion of the cameras on Europa Clipper and their ability to detect changes in surface properties as described here would be helpful.

We referenced the EIS and MISE instruments and gave MISE's spatial resolution and purpose (detection and mapping of salts, acids, organics). Whether these instruments and their synergy will be able to detect the specific signatures obtained in this study would require further modelling and that work has to be conducted in collaboration with the instrument teams, as per Europa Clipper's Rules of the Road.

11. Line 326. What is the resolution of the thermal anomalies detectable by E-THEMIS? Can it see anomalies at the ~1 K level over the spatial scales needed here?

We added E-THEMIS's expected temperature precision (1.2 K) in the discussion.

12. Lines 448–449. The paper states that radial heat transport is not significant because the reservoir boundary is nearly isothermal. But I would think that would mean that lateral heat transport is small. Radial heat transport would be into and out of the reservoir? And that could still be important?

We meant here that for a spheroid (lens)-shaped reservoir, most of the heat is exchanged through the horizontal-oriented walls, and little by the "sides", as sills are much wider than they are tall. I rephrased to clarify: "This is a good approximation, with only small uncertainties introduced through *de minimis* heat transport along the thin, nearly isothermal boundary normal to the horizontal plane."

13. Line 446. The word "is" should be omitted.

Done

14. Lines 476–481. The parameter values all appear reasonable, but some references should be given. Perhaps to put these in a table would be the easiest way to present this?

We added a reference to the IAPWS (see <https://www.iapws.org/>) as these are standard / well accepted properties of liquid water.

Reviewer #1 (Remarks on code availability):

I was able to access the code, but did not actually test it. The README file appears to be empty.

To the popular demand –for good reasons– the README file is now written and describes how to use the code.

Reviewer #2 (Remarks to the Author):

Key results:

The authors report on a modeling study that investigates the storage, physicochemical evolution, and eruption of brines contained within discrete reservoirs within Europa's ice shell. This work builds on previous work by the authors, and takes the important step of combining focus on the freezing reservoir itself with assessment of the composition of cryolavas that would be erupted, and the effect of the presence of the brine reservoir on the thickness of the ice shell. The authors present results for reservoir longevity, surface temperature anomalies, ice shell thickness variations, and total erupted volumes for three plausible ocean compositions, as well as pure water. They show that predicted erupted compositions bear close similarity to the initial reservoir/ocean composition, with increasing salinity over time as the reservoirs freeze, which would allow for discrimination between eruptions sourced from a subsurface reservoir vs. the ocean. As such, this is a very comprehensive treatment that provides predictions that can be tested with a range of instruments flying on the upcoming missions to the Europa system.

Significance:

The work provides a set of predictions to inform upcoming missions to the Jupiter System that will allow for identification of past or present fluid reservoirs and their eruptions. Identification of locations hosting liquid water is highly significant for the outer solar system exploration objective of searching for habitable environments.

Data, methodology, validity of approach:

As we await new data sets from upcoming missions to Europa, we rely heavily on theoretical modeling, in this case modeling of the physicochemical evolution and consequences of brine reservoirs in the ice shell. The description of the modeling components, in both the main text and supplemental material, is sufficiently detailed, and the approach adopted seems to be robust. The strength of the present work lies in combining different model components (conservation of heat through the ice shell in response to the freezing of the brine reservoir, chemical evolution of the reservoir, pressurization and eruption of the freezing reservoir, and stress response of the ice shell), in order to produce a more holistic evaluation of the dynamics of brine reservoirs in the ice shell. The authors explore an appropriately broad parameter space, which captures the plausible (but as yet unknown) range of conditions that may exist. Figure 3 provides an excellent summary of the results of all simulations. While we still don't know how such liquid reservoirs might be emplaced in the first place (which the authors acknowledge), I concur that the conclusions regarding reservoir and eruptive evolution are valid and well supported by the modeling.

This is a modeling investigation that uses previously published data as model parameters and constraints. As such, those data have already been validated. The code underpinning this work is provided on GitHub, but has not been test-run by this reviewer.

Clarity and context: The manuscript was a pleasure to read. It is well structured and clearly written, and appropriately discusses previous work as context for the new results presented.

Suggested improvements:

The quality of the manuscript is high and I have only minor comments, marked up on the manuscript and supplemental material files. In addition to typo, formatting, and wording comments left solely in the marked-up files, I reiterate a few comments (all minor) here:

Thank you very much the very helpful comments in the PDF, we addressed them all and corrected typos.

– Line 130. “phase-energy” I assume you mean phase-change energy here (i.e. latent heat)?

Changed for “latent heat”

– Line 132. If I'm reading figure 1c correctly, it looks like ~3 m thinning, not 30 m.

Good catch, thank you!

– Line 220. Add “at the surface” after “modified” for clarity

Done

– Line 280. Regarding “... because the thermal gradients driving heat advection are lower...”, do you mean heat conduction through the ice, or not just heat conduction, but also advection due to ice convection? In either case, conduction is relevant so should be included.

We changed “advection” for “conduction”.

– Line 295. I agree that ocean-derived liquids would like differ in composition than liquids derived from in situ heating. But what if the locally melted ice was salty? Chivers et al. have shown that complex compositional layering can be produced in subsurface ice, so the compositiona; doistinction between ocean-derived and locally-derived (in-situ) fluids might not be as clear cut as briny vs. pure water fluids.

Here we intended to communicate that in-situ melted reservoirs would be enriched in low-eutectic salts compared to material coming from the ocean (understand any of the three input compositions tested), which I clarified in the next sentence: “In situ melting of eutectic mixtures would refreeze near the eutectic, and the material stored in the reservoir and erupted at the surface would thus be relatively enriched in lower-eutectic temperature salts such as chlorides compared to the oceanic compositions tested.”

– Line 424. Last line of paragraph is incomplete.

Fixed, thank you.

– Line 443-445. Description of reservoir geometry: It's hard to envision what you mean here. An annulus is the region between two concentric circles... ring-shaped like a washer. I don't see how a spherical reservoir can be represented by an annulus. Do you simply mean that the reservoir is represented in 2D by a circle rather than an annulus? A clearer explanation (or a figure) is needed to clarify the geometry.

We tried to clarify by rephrasing this paragraph: “Because of our 1D approximation for the model grid, the liquid reservoir does not technically own a horizontal extent, but is rather described by a global layer of water of imposed initial thickness. However, we use a spherical description to best simulate energy transfer in and around the reservoir.”

– Line 479. Ra is dimensionless, so here do you mean that to calculate Ra you need to multiply by some temperature and the dimension of the reservoir? Please clarify by providing explicit explanation of K and D. Or perhaps give the Rayleigh number as $Ra = 5 \times 10^{10}(\Delta T D^3)$, where ΔT is... and D is...

This was indeed unnecessarily complicated and unclear. We condensed the two sentences into one and simply gave the final Ra.

References.

The referencing is quite comprehensive but the authors might consider whether they would like to cite the following recent papers, which also bear upon the issues discussed in the paper:

- Chivers et al. (2023) Stable brine layers beneath Europa's chaos, PSJ
- Carnahan et al. (2022) Surface-to-ocean exchange by the sinking of impact-generated melt chambers on Europa, GRL
- Hesse et al. (2022) Downward oxidant transport through Europa's ice shell by density-driven brine percolation, GRL
- Winkenstern and Saur (2023) Detectability of local water reservoirs in Europa's surface later under consideration of coupled induction, JGR Planets

Thank you for the great suggestions, we added these references in the introductory paragraphs.

Reviewer #2 (Remarks on code availability):

I did view the GitHub site, but did not have sufficient time to dedicate to figuring out how to run the code. A more detailed readme file would be beneficial.

To the popular demand –for good reasons– the README file is now written and describes how to use the code.

Reviewer #3 (Remarks to the Author):

This paper addresses an important topic for future explorations of Europa's habitability with NASA's Europa Clipper and ESA's JUICE missions, offering a comprehensive approach to distinguish the source reservoir of erupted water. The manuscript is generally well-written, and I hope my comments will help improve the clarity of the paper.

1. Lines 168-172: This is a reasonable assumption, given experimental data for terrestrial seawater. You could also mention that this assumption has been used in existing Europa literature, such as Zolotov & Shock (2001, JGR, 106, 32895-32827). We added "This assumption is consistent with experimental data for terrestrial seawater and previous Europa literature (Zolotov and Shock, 2001)"

2. Ocean Composition Table: It would be helpful for readers to include a table describing the assumed ocean composition (i.e., key ions, dissolved volatiles, pH) used in these calculations, even though this information is cited from [40]. A brief description of the major differences in these cases would also be appreciated, as it's not immediately clear which comet compositions were adopted (e.g., is it from comet 67P or an average composition of multiple comets?).

Yes, Melwani Daswani et al. (2021) used 67P for cometary material composition. We are reaching the length limit for the paper, but we referred to the methods where we added more detail on the three different compositions and a table with the input molalities.

3. Figure 2a & 2c: Fig. 2a seems to omit the evolution of Si, and Fig. 2c refers to a mix of CI and CM with added cometary materials, but S and Mg appear to be missing. Without further explanation, it's not obvious why Si and Ca differ by an order of magnitude between Fig. 2b and 2c, given that Fig. 2c is a mixture of materials from CI and CM, with some cometary input.

The MC-Scale model used Monte Carlo simulations where the probability of accreting different types of carbonaceous chondrites (CI, CM, CV, CK, CR, and CO) was weighted by their formation distance from Jupiter. Cometary material was then added to compensate for insufficient volatiles produced by this accretion scenario. This fundamentally different formation model, rather than late cometary addition alone, explains the order-of-magnitude differences in elemental concentrations like Si and Ca between scenarios. We clarified in the methods. Additionally, very low concentrations were approximated to 0 in the compositional data to allow the PHREEQC freezing runs to proceed to the lowest possible temperatures without convergence issues at very high ionic strengths (Naseem et al., 2023). We clarified this in Fig. 2 caption.

4. Volatile Exsolution: While volatile exsolution isn't the main focus of this paper, it's been discussed in ocean worlds literature as a potential driver for eruptions (e.g., Mitchell et al. 2024, <https://doi.org/10.1029/2023JE007977>). Within the context of this model, it might be useful to discuss to what extent this process might contribute to the eruption process.

Volatile exsolution might happen once brines reach the surface, as we discussed in third-to-last paragraph. The exsolution of volatiles as proposed in Mitchell et al. (2024) cannot be the driving force of eruptions on Europa where subsurface pressure is required to maintain a cryovolcanic conduit open, contrary to Enceladus where tides can be sufficient. We thus don't expect this process to impact any of our results.

5. Thermal Implications (Line 227): Given the maximum increase of 1.5K for the shallowest reservoirs, what are the implications for thermal measurements in identifying hotspots? Does this explain why telescopic measurements have not reliably detected endogenic heat? (e.g., <https://iopscience.iop.org/article/10.3847/1538-3881/aa8769>) We added E-THEMIS's temperature precision (1.2 K) in the discussion to clarify its relevance. Telescopic measurements would not have the spatial resolution to identify the hot spots we are describing here.

6. Testing Models with Current and Future Missions: The paper provides a strong discussion of how future missions could test these models. However, is there an opportunity for current or near-future Juno observations to help? While the discussion focuses on spectral, E-THEMIS, and REASON instruments, it's unclear how future gravity data could help distinguish between water erupted from deep ocean sources versus shallow liquid reservoirs, as mentioned earlier in the abstract.

We added the following sentence to clarify: "Finally, the density changes associated with the presence of a large-scale liquid reservoir, and an additional local ice shell thinning, might result in a gravity anomaly potentially detectable by gravity measurements (Mazarico et al., 2024)".

Regarding Juno, observations from the IR spectrometer JIRAM and the microwave radiometer MWR may be influenced to some degree by the processes described here (see Filacchione et al., 2019 and Zhang et al., 2023). However, signatures of reservoir emplacement and cryovolcanism are not likely the first-order drivers of these highly spatially integrated observations, which makes them less relevant to discuss here.

Reviewer #3 (Remarks on code availability):

The code appears to be organized, but the README file does not provide any instructions for installing and running the application

To the popular demand –for good reasons– the README file is now written and describes how to use the code.

Response to the reviewers' comments:

The authors thank the three reviewers very much for their time and contributions that greatly improved this manuscript.

Reviewer #1 (Remarks to the Author):

The authors have responded to all my comments on the original submission, and I'm ready to recommend this for publication. I have just a couple of further recommendations, which can be easily handled during the proofs stage; there is certainly no need for another round of review.

Regarding my original point 3, the authors explanation seems reasonable, but it would be good to include a mention of these complicating mechanisms (e.g., pressurization, expansion, vaporization), note that their not being included, and a brief (one sentence) explanation of why these can be safely ignored at this stage.

We added the following sentence to put the erupted volume in perspective: “This is consistent with what we expect for the assumptions we made: (i) the reservoir is disconnected from the ocean with no water refilling, (ii) the frozen ice layer has no porosity, keeping the water from infiltrating the ice shell, and (iii) the cryomagma is maintained at a pressure high enough that it does not volatilize before it reaches the surface.”

Note that we do not address how much of the erupted volume is volatilized after eruption, as it is beyond the capabilities of our simulation, but this is amply discussed in the discussion (see “Implication for future missions”).

Regarding my original point 10, I see the authors cite the MISE paper in SSR, but not the corresponding EIS paper (Turtle et al., 2024). That one is now published online: <https://link.springer.com/article/10.1007/s11214-024-01115-9> and should surely be included here.

Thank you for catching this, we added the citation.

Reviewer #1 (Remarks on code availability):

I have not actually tried using the code, but have had no problems accessing it. The README and License are complete now.

Reviewer #3 (Remarks to the Author):

I have carefully reviewed the revised manuscript and the additional information provided by the authors. I am satisfied that my comments and concerns have been adequately addressed. The revisions improve the clarity and robustness of the study. I recommend the manuscript for publication in its current form.